# Atomic-Scale Tracking of Dynamic Nucleation and Growth of an Interfacial Lead Nanodroplet

**DOI:** 10.3390/molecules27154877

**Published:** 2022-07-30

**Authors:** Xiaoxue Chang, Chunhao Sun, Leguan Ran, Ran Cai, Ruiwen Shao

**Affiliations:** 1Analysis & Testing Center, Beijing Institute of Technology, Beijing 102488, China; cxx115@bit.edu.cn; 2School of Environmental and Safety Engineering, North University of China, Taiyuan 030051, China; chunhaosun9808@163.com; 3School of Medical Technology, Beijing Institute of Technology, Beijing 100081, China; ranleguan@163.com

**Keywords:** nucleation, crystal growth, in situ observation, aberration-corrected transmission electron microscopy

## Abstract

Revealing the evolutional pathway of the nucleation and crystallization of nanostructures at the atomic scale is crucial for understanding the complex growth mechanisms at the early stage of new substances and spices. Real-time discrimination of the atomic mechanism of a nanodroplet transition is still a formidable challenge. Here, taking advantage of the high temporal and spatial resolution of transmission electron microscopy, the detailed growth pathway of Pb nanodroplets at the early stage of nucleation was directly observed by employing electron beams to induce the nucleation, growth, and fusion process of Pb nanodroplets based on PbTiO_3_ nanowires. Before the nucleation of Pb nanoparticles, the atoms began to precipitate when they were irradiated by electrons, forming a local crystal structure, and then rapidly and completely crystallized. Small nanodroplets maintain high activity and high density and gradually grow and merge into stable crystals. The whole process was recorded and imaged by HRTEM in real time. The growth of Pb nanodroplets advanced through the classical path and instantaneous droplet coalescence. These results provide an atomic-scale insight on the dynamic process of solid/solid interface, which has implications in thin-film growth and advanced nanomanufacturing.

## 1. Introduction

Research on the growth mechanism of nanostructures is an important issue in chemistry and materials science, guiding the creation and design of new substances and spices [1,2]. Generally speaking, the growth of nanostructures begins with the initial critical nuclei and the subsequent crystallization evolution [3,4]. In these two steps, the subtle discrepancy in the initial nucleation stage induces the subsequent formation pathway and growth route of the phase structure. Nucleation is the critical point of multiple atoms from disorder to order and the crucial symbol of the subsequent growth and advance of the materials [5,6]. The nucleation process of solid-state phase transition is relatively slow in which phase transition can be suspended after the reaction stops. The static characteristics of nucleation based on the solid state have been widely explored by researchers [7,8]. As a comparison, the dynamic nucleation of nanodroplet origins in the coalescence and growth from the rapid interface contact changes, resulting in the fleet phase reaction. The crystallization of nanostructures follows behind nuclei formation, according to the classical homogeneous nucleation and growth mechanism. The amount of atoms in the liquid phase spontaneously and randomly gather to form a cluster, which further transitions to the nuclei. Once nucleated, more and more atoms or molecules can gradually adhere to the nuclei and grow and finally form a stable crystal structure. However, it is noted that not all nucleation and growths follow the classical mechanism [3,9]. Many results show the existence of other parallel ways on the classical nucleation pathway, such as particle coalescence or attachment-mediated nucleation, phase transition-induced two-step nucleation, etc. [10,11].

Limited by the hardware conditions of real-time observation, the direct evidence of the nucleation and crystallization of liquid–solid-phase transition was still lacking, despite its great significance in theory and practical applications [12,13]. Therefore, real-time observations of the nucleation and crystallization of liquid–solid-phase transitions is the focus of the comprehensive exploration of the liquid-phase particles transition process. Moreover, the liquid nucleation process at the interface of the liquid–solid base is significantly different from the crystallization process in the pure liquid, as demonstration by the amount of available simulation results. In order to observe the critical reaction phase in real time, it is necessary to select a precise experimental technology that not only realizes the imaging of the initialization of the nanodroplets but also displays the dynamic process of nucleation.

Currently, in situ dynamic experimental observation was used to accurately capture the multistep nucleation stage that deviates from the classical nucleation path. Although such observations put forward higher requirements for experimenters, more in-depth and detailed results can be obtained at the atomic level. Many of the studies provided implied that they conceived of atomic simulations for the fluctuation of the solid–liquid interface [14,15,16]. With the advance of transmission electron microscope (TEM) down to the atomic scale, we can directly observe atom aggregation, crystal nucleation, and crystallization growth at a higher resolution. Taking advantage of the high-temporal and spatial resolution, in situ TEM equipped with Cs correction is the best choice for dynamic and real-time observations of the evolutions of nanostructures [4,17]. The resolution of the spherical aberration correction transmission microscope can reach 0.6 Å, which is significantly higher than that of the ordinary electron microscope. For example, Alivisatos et al. used high-resolution TEM to reveal the distinct nucleation and growth mechanism of platinum nanocrystals [18]. They created their own in situ reaction system in which an electron beam was used to initiate nucleation, revealing the growth of platinum nanocrystals through a monomer attachment or particle aggregation in the solution [19,20]. The reason why the dynamics of droplet coalescence are widely noticed is that they appear in a variety of physical, chemical, and biological phenomena [21,22]. The progress of micro-cognition has a significant effect on the macro-application, which makes this phenomenon of great significance for experimental and theoretical research [23,24,25,26].

In this work, we irradiated PbTiO_3_ nanowires to generate Pb nanoparticles with the electron beam in a TEM equipped with spherical aberration correction and conducted in situ atomic-scale dynamic observations. Although the melting point of bulk lead is 600.6 K, the melting temperature of Pb nanoparticles can be decrease to room temperature due to the size effect [27,28]. PbTiO_3_ is a typical perovskite ferroelectric, in which Pb atoms will be gradually motivated from the interior of nanowires under the irradiation of an electron beam. Therefore, thin PTO nanowires can be used as an ideal base to explore the nucleation pathway and crystallization mechanism of Pb nanoparticles. The nucleation pathways and crystallization route of Pb nanoparticles help to enrich the nucleation theory and guide the growth and synthesis of crystalline, amorphous, and cluster materials.

## 2. Results and Discussion

The time-resolved HRTEM images of Pb nanodroplets growth under an electron dose of about 2 nA (dose 11,000 e/Å^2^s) are displayed in Figure 1, revealing the two-step nucleation pathway of Pb nanodroplets based on the PTO nanowire. It should be noted that the intermediate state formed within a few seconds during the focusing process. When a clear image was taken for the first time, the Pb nanodroplets were precipitated on one side of the nanowire, and the opportunity to observe the initial state of the material was lost. Fortunately, no obvious Pb precipitation formed in the region of the nanowire that was not irradiated by the electron beam (Figure 1a). Such a target region can be an ideal platform to investigate the initial stage of nucleation and the subsequent crystal growth (Figure 1a–f).

The surface of the pristine PTO nanowire is pure, and the d space of (200) is 0.27 nm. The Pb atoms originally existed in the PTO structure in the form of oxide. After 2.5 s of continuous electron beam irradiation (Figure 1a), during the initial prenucleation stage, many ultra-small drop-like circular precursors (<3 nm) formed inside or on the surface of the PTO. This Pb droplet then fluctuated between the cluster and the crystal. With further electron beam irradiation, atomic rearrangement took place as the surgency of the Pb atoms. This Pb droplet then fluctuated between the cluster and the crystal (Figure 1b).

Due to the lattice-induced effect, the nucleation could be completed in a very short time and used as the critical nucleus to grow on the matrix [29]. The initial Pb was tiny, only on a nanometer scale. The low melting point induced by the size effect triggered the initial Pb to form droplets. The newly generated small-sized Pb droplets (the precipitation inside the nanowires is shown in the red coil in Figure 1a–c) grew and migrated on the PTO nanowire surface. 

We also observed the coalescence process of small Pb droplets (labeled 1 and 2 in Figure 1c). The unstable nanoparticles gradually grew up and coalesced with each other in a short time, transforming into an internal locally ordered structure (Figure 1c–e). At 14.75 s, droplet 1 and droplet 2 coexisted on the surface of the nanowires and then further grew by fusion with each other. After the fusion of the two nanodroplets, the total surface area decreased, and the structure fluctuated between the droplets and the crystal. The reduction of the surface energy and bonding rearrangement led to a local temperature rise in the newly formed nanograins and helped to form larger-sized and multi-faceted balanced nanocrystals and stabilized the crystal structure. The Fast Fourier Transform (FFT) diffraction pattern of Pb nanoparticles in Figure 1g further confirmed the structural transformation from the local ordered structure in the droplet to a stable crystal. After fusion, the crystal area became larger, and the crystallinity was better. According to the local FFT results, only the grain orientation changed during the whole process, and no phase transformation occurred.

In order to further reveal the growth and fusion process of nanocrystals and explore the influence of size factors on the fusion mechanism, we further compared the differences in the fusion process of different sizes of grains. Figure 2a–c show the coalescence process from a small crystal (~6 nm) to another large nanocrystal (~15 nm). The small Pb nanocrystals migrated to the surface under electron irradiation, and the structure fluctuated continuously. Due to its own instability, the small crystal coalesced with the large nanocrystals rapidly, and the atoms rearranged into large crystals. This process was completed within 4 s. The coalescence process of several small nanocrystals was similar to the coalescence between the large nanocrystal and the small nanocrystal, following a rapid step-by-step migration mechanism. Figure 2d–f show the coalescence process of three nanocrystals (~10 nm), which migrated on the PTO nanowire and fused in turn. The whole fusion process took less than 3 s. A combined small plane structure with double junctions or triple crystal boundaries formed during the rearrangement of the atoms, which was quite different from the case in Figure 2a–c.

With the increase of the size of the nanocrystals, the crystal orientations tended to be maintained during the coalescence process. Compared with the continuous jump transformation of small crystals, the frequency and speed of the morphological changes and grain movement of large crystals are greatly reduced. The speed of atomic migration and rearrangement between large nanocrystals are relatively slow. Figure 2g–i show the coalescence process of two large grains with sizes of about 15 nm. It can be seen that the grain orientation of the two crystals changes during the fusion process, and the atomic rearrangement begins after the initial contact. However, an interface or grain boundary forms after the contact of the two grains, and the atoms will migrate and arrange along the grain boundary with different orientations.

To study the dynamic change of the interface after grain fusion, we selected two typical clear fusion interfaces for observation. Figure 3a–f show the dynamic change process of the interface after the fusion of two large grains with a size of about 15 nm in Figure 2h–j. It can be seen that the grain boundary expands and transforms into twinning structures during the grain fusion process. The sliding process of the twinning structures was observed between 1.5 s and 3.75 s. Orange arrows point to regions where the atomic arrangement changes the most. Different from the stable interfaces in Figure 3a–f, the internal activities of the same-sized grains in Figure 3g–i are still frequent, and the interface also disappears rapidly during the activity, and the fusion transforms into a complete single crystal. In order to reduce the surface energy, the surface area of the droplets tends to shrink automatically, and there is a driving force for fusion between the droplets. Atomic rearrangement occurs when small droplets spontaneously fuse with each other or merge into larger droplets, and the redistribution of atoms in nanodroplets reduces the surface energy.

Atomic rearrangement is very common in small nanocrystals. Compared with the rapid atomic migration and rearrangement of small particles, the energy required for the overall orientation change of large nanocrystals is greater. It is observed that the slow atomic migration and rearrangement rate will not change the atomic arrangement frequently. The influence of the initial orientation difference between small nanoparticles and large nanoparticles in the fusion can be ignored. After the final fusion, the small grains tend to integrate into the large nanoparticles, change the original orientation to adapt to the large nanoparticles, and form a new nanocrystalline orientation under its influence. It leads to the formation of relatively stable grain boundary particles after contact.

In order to uniformly analyze the structural changes of the Pb nanocrystals observed in all micro individuals, we identified and summarized the changes of the area (Figure 4a) and shape (hemispherical, triangular, trapezoidal, pentagon, and hexagon); the changes of the area (Figure 4b) with time; and the changes of the length (Figure 4c) and shape and used the scatter diagram to reflect the statistical results (original detailed data of the statistical results can be seen in Appendix A). The orange curve shows the time-dependent curve of the average area of the nanocrystals. Examples of actual crystal profiles corresponding to different simplified symbols are shown in Figure 4d. In the first stage, it transforms among semicircles, triangles, trapezoids, and pentagons. When the diameter of the Pb nanocrystals exceeds ~5 nm, the triangle disappears, and the structure changes between hemispherical, trapezoidal, pentagonal, and hexagonal (Figure 4c). The mean and standard deviation area for the area and length of each shape are listed in Table 1, in which it can be seen that the most common in large sizes are pentagons and hexagons. When the length exceeds 7 nm, it gradually turns into a trapezoid or pentagon. After 80 s of growth and atomic rearrangement, the shapes of nanocrystals gradually change from hemispherical structures to stable and regular polyhedral structures (pentagonal and trapezoidal structures) with oxide support. Subsequently, the polyhedral nanocrystals either merge with the small nanocrystals through the step migration mechanism or coalesce with another large polyhedral nanocrystal through atomic rearrangement to reduce the surface energy and form a more stable epitaxial structure.

## 3. Materials and Methods

Sample preparation. In this experiments, high-quality PbTiO_3_ nanowires were used as the source of Pb particle precipitation and the support of the whole reaction process. The PbTiO_3_ samples were ultrasonically dispersed in absolute ethanol for 30 min, and then, a few drops of this dispersion were placed on an ultrathin carbon film, which was subsequently dried under an infrared lamp for the in situ TEM experiments.

HRTEM observation and data analysis. Pb nanocrystals were nucleated by irradiating PbTiO_3_ nanowires with a low-dose electron beam in a spherical aberration correction TEM (Thermo Fisher Scientific Company Titan Themis Z) equipped with an X-FEG gun, which was also used for atomic structure imaging of the Pb nanoparticles. The screen current was about 2 nA (dose 11,000 e/Å^2^ s). The TEM acceleration voltage for spherical aberration correction was 300 kV. The camera that recorded the real-time observation video of the experiment was a CCD camera (Gatan 832). The FFT patterns and inverse FFT were created with Digital Micrograph (Gatan) software. 

## 4. Conclusions

This study not only provided a promising method for manufacturing nanodroplets on PTO one-dimensional nanowires but also proved a new method for observing and recording the whole process of PTO nanowires forming Pb nanocrystals under low-electron beam irradiation. The ultrahigh resolution transmission electron microscope with spherical aberration correction can provide experimental direct observational support for the whole process of nanocrystal formation, from nucleation to growth and fusion at the atomic scale, helping to understand the development mechanism of each stage of nanocrystal nucleation and show the microscopic details of the rapid changes. Based on the experimental phenomena, the growth and evolution of Pb nanodroplets mainly follow three stages: (I) the formation of atomic Pb clusters or droplets, (II) the critical nuclei of nanodroplets are formed indirectly by adding atoms directly through the classical way, (III) promoting the growth of nanocrystals by adding atoms and droplet coalescence. In the nucleation process, both locally ordered clusters and amorphous droplets can coexist as intermediate states, which indicates that the growth process of interfacial Pb nanodroplets is very complex. The whole process of Pb crystal formation can greatly help us understand the growth mechanism of liquid nucleation of microcrystals and promote the expansion of nanoscale nucleation cognition.

## Figures and Tables

**Figure 1 molecules-27-04877-f001:**
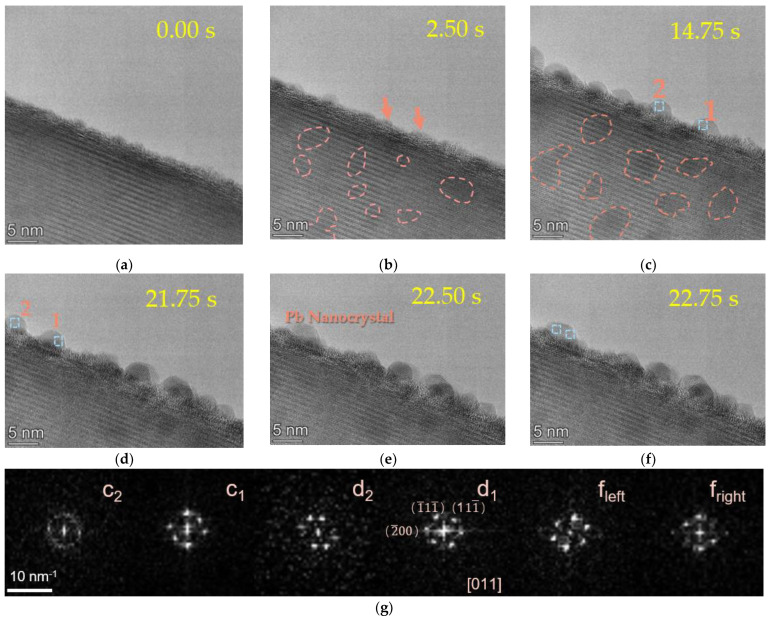
The time-lapse HRTEM (high-resolution transmission electron microscope) images of the two-step nucleation and growth path of Pb droplet crystal under electron beam irradiation. (**a**–**c**) Pb nanocrystalline particles precipitate (0 s), nucleate (2.5 s), and grow up (14.75 s). The traces of Pb precipitation and growth inside the nanowires were circled by red coils. (**b**) The red arrow shows the initial state of the two particles. (**d**,**e**) The bonding process of two small crystals. (**f**) The HRTEM image shows the shape change of the newly formed crystal and the structural fluctuation after fusion. (**g**) The corresponding fast Fourier transform (FFT) of the labeled nanoparticle areas with the blue boxes.

**Figure 2 molecules-27-04877-f002:**
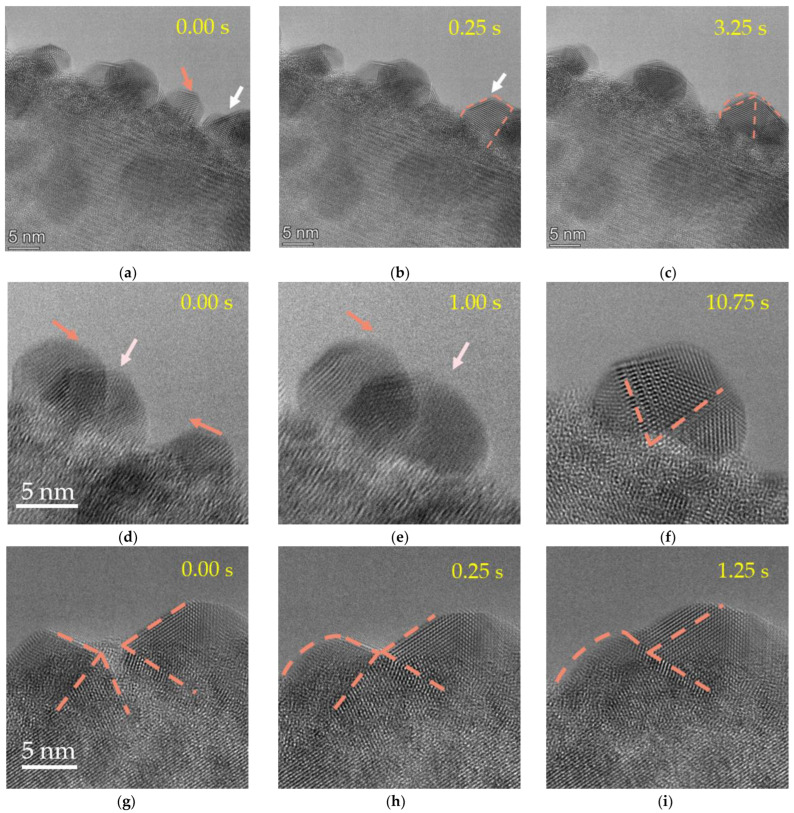
The time-lapse HRTEM images of Pb droplets coalescence and growth of different sizes. The orange arrows point to the particles that are about to move and indicate the direction of their movement, and the white arrows are the particles that are subject to fusion. (**a**–**c**) The coalescence process of a small nanocrystal and a large nanocrystal. (**d**–**f**) The three small crystal coalescence and atomic rearrangement processes. (**g**) The coalescence process of two large independent particles, and the different orientation boundaries of the crystal plane are drawn with red auxiliary lines, respectively. (**h**) The change of the crystal orientation at the moment when the two grains fuse. (**i**) The change of the crystal orientation after the two grains fuse.

**Figure 3 molecules-27-04877-f003:**
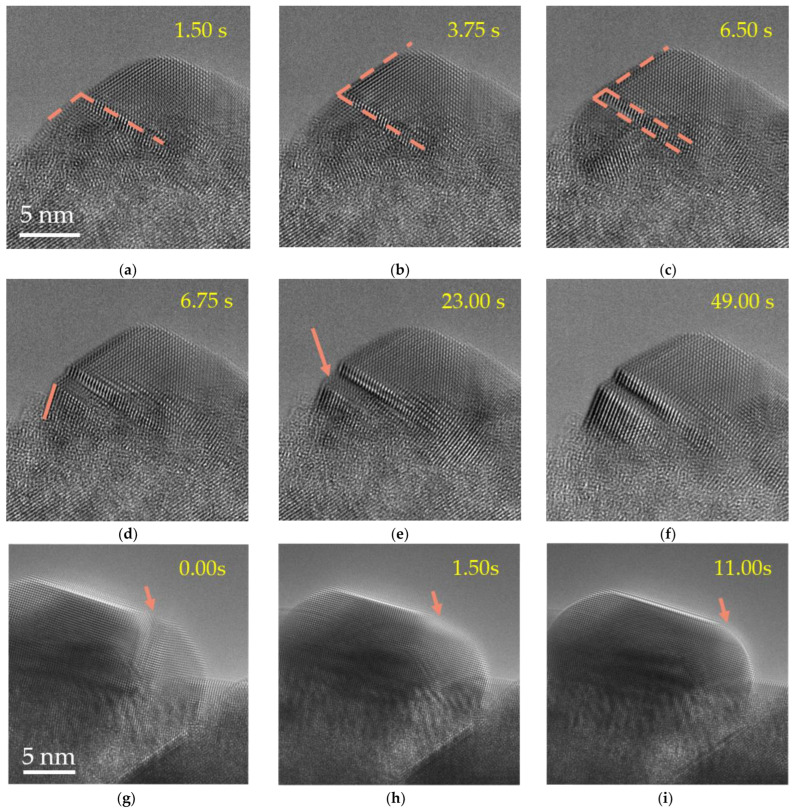
The time-lapse HRTEM images of the fusion process of two Pb particles under electron beam irradiation. (**a**–**c**) The changes of the surface and interface after fusion are marked with a red outline. (**d**–**f**) The dynamic process of the interface orientation arrangement and change. (**g**–**i**) The process of interface disappearance and atom rearrangement in the fused grains. Orange arrows point to regions where the atomic arrangement changes the most.

**Figure 4 molecules-27-04877-f004:**
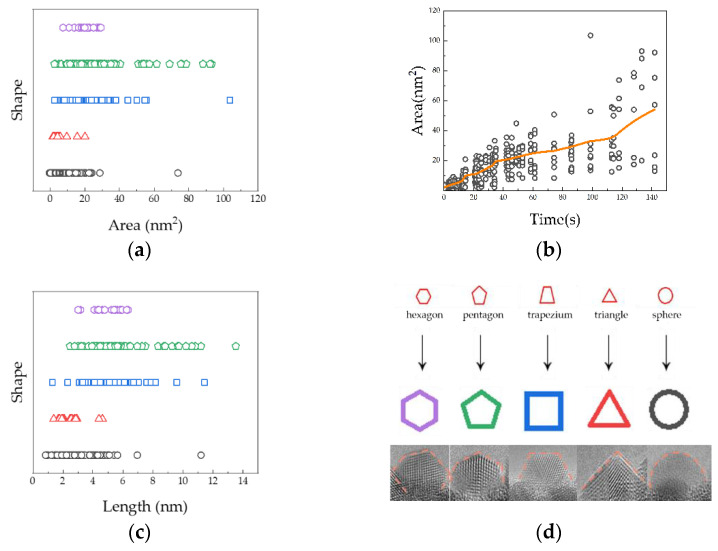
The grain parameters such as length, width, size, and area with time growth are counted and the correlation between the parameters and the change of the grain shape. (**a**) The relationship between the area and grain shape. (**b**) The relationship between the time and area. (**c**) The relationship between the length and shape. (**d**) The corresponding example of the symbol and actual shape.

**Table 1 molecules-27-04877-t001:** The mean and standard deviation area for the area and length of each shape.

Title 1	Hemispherical	Triangular	Trapezoidal	Pentagon	Hexagon
Area mean (nm^2^)	3.00	2.46	4.88	5.80	5.01
Area standard deviation area	±1.19	±0.95	±1.20	±1.39	±0.97
Length mean (nm)	8.47	5.63	19.80	28.73	20.45
Length standard deviation area	±2.92	±2.25	±3.65	±4.43	±2.41

## Data Availability

Not applicable.

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
