# Peer review of "Atomic-Scale Tracking of Dynamic Nucleation and Growth of an Interfacial Lead Nanodroplet"

_molecules, 2022, doi:10.3390/molecules27154877_

Round 1
Reviewer 1 Report
This article discusses work involving the growth pathway of Pb nanodroplets created during electron irradiation. Interesting work, but I have a few comments.
1. In the Materials and Methods section, please mention the irradiation conditions such as dose.
2. In Figures 2 and 3, please indicate what the arrows are in either the figures or the caption.
3. I do not understand the purpose of Figures 4(a) and (c). It might be better to provide a table which lists the mean and standard deviation area for each shape.
4. In Figure 4(b), it is suggested to provide a trendline to the data.
Reviewer 2 Report
The paper by Chang et al. deals with atomic-scale tracking of dynamic nucleation and growth of an interfacial lead nanodroplet. The paper is well written and the scientifical results seems sound. I will recommend publication, after the authors will perform minor revision of their manuscript. The following points should be addressed by the authors, before the manuscript may be reconsidered for publication:
1. The abbreviation – Fast Fourier Transform (FFT) is introduced twice. In the Section 4. Materials and Methods as well as in the caption of Figure 1 (g). This abbreviation (FFT) should be introduced only once in the paper under review.
2. In the Reference Section, in the Reference [28], in the abbreviation of journal title: Mater. Sci. Eng., R – should be – Mater. Sci. Eng.: R. Namely, instead of comma after Eng., - should be used the – Colon (punctuation). Moreover, the abbreviation of the journal title in the Reference [28] should be in italic – Mater. Sci. Eng.: R.
3. In the Introduction of the paper under review, it is necessary to mention also some recent paper dealing with the theoretical ab initio investigations of PbTiO3 perovskite material bulk and its (001) nano-surfaces, for example [1]:
Reference:
[1] R. I. Eglitis, J. Purans and R. Jia, Comparative hybrid Hartree-Fock-DFT calculations of WO2-terminated cubic WO3 as well as SrTiO3, BaTiO3, PbTiO3 and CaTiO3 (001) surfaces, Crystals 11, 455 (2021).
